# An IoT- and Cloud-Based E-Waste Management System for Resource Reclamation with a Data-Driven Decision-Making Process

**Mithila Farjana** , **Abu Bakar Fahad** , **Syed Eftasum Alam** **and Md. Motaharul Islam** *

Deptement of Computer Science and Engineering, United International University, Dhaka 1212, Bangladesh; mfarjana201127@bscse.uiu.ac.bd (M.F.); afahad201119@bscse.uiu.ac.bd (A.B.F.); salam201133@bscse.uiu.ac.bd (S.E.A.)
* Correspondence: motaharul@cse.uiu.ac.bd

**Abstract:** IoT-based smart e-waste management is an emerging field that combines technology and environmental sustainability. E-waste is a growing problem worldwide, as discarded electronics can have negative impacts on the environment and public health. In this paper, we have proposed a smart e-waste management system. This system uses IoT devices and sensors to monitor and manage the collection, sorting, and disposal of e-waste. The IoT devices in this system are typically embedded with sensors that can detect and monitor the amount of e-waste in a given area. These sensors can provide real-time data on e-waste, which can then be used to optimize collection and disposal processes. E-waste is like an asset to us in most cases; as it is recyclable, using it in an efficient manner would be a perk. By employing machine learning to distinguish e-waste, we can contribute to separating metallic and plastic components, the utilization of pyrolysis to transform plastic waste into bio-fuel, coupled with the generation of bio-char as a by-product, and the repurposing of metallic portions for the development of solar batteries. We can optimize its use and also minimize its environmental impact; it presents a promising avenue for sustainable waste management and resource recovery. Our proposed system also uses cloud-based platforms to help analyze patterns and trends in the data. The Autoregressive Integrated Moving Average, a statistical method used in the cloud, can provide insights into future garbage levels, which can be useful for optimizing waste collection schedules and improving the overall process.

**Keywords:** IoT; cloud; e-waste; pyrolysis; Generative Adversarial Networks; bio-fuel; recycling

## 1. Introduction

E-waste refers to repudiated electronic devices, such as computers, mobile phones and other electronic equipment, that are at the verge of their efficacious use. Owing to the unrelenting momentum of technological innovation, a growing multitude of individuals are procuring electronic devices with regularity; thus, this begets roughly 54 to 60 million metric tons of e-waste every year, averaging some 7 kg of e-waste per capita. Pursuant to the Global E-waste Statistics Partnership, this is expected to increase to 74.7 Mt by 2030. By 2025, it is estimated that Asia will generate the highest volume of e-waste, at 24.4 million metric tons, followed by the Americas (13.4 million metric tons) and Europe (12.8 million metric tons). Scarcely around 15 percent of global e-waste was collected and recycled in 2014, with the remaining 85 percent being discarded in landfills or incinerated [1].

This situation gives rise to a profound disquietude and engenders a palpable sense of apprehension. It is incumbent upon us to take substantive action. The deleterious effects of electronic waste on the environment are manifold and unequivocal. It has been empirically demonstrated that the materials utilized in the construction of these devices, when containing high concentrations of lead and mercury, are capable of perniciously poisoning the surrounding soil in landfills. Once discarded, the components of e-waste

become veritable toxins for the ecosystem, gradually seeping into the earth and causing damage to the atmosphere [2]. This process releases noxious chemicals into the air, thereby exacerbating air pollution. Furthermore, as these toxic materials are carried by rainwater or groundwater, they can affect both terrestrial and aquatic wildlife, rendering e-waste an omnipresent threat to environmental health. The identification and separation of e-waste from municipal solid waste (MSW) is a challenging task that requires significant resources. Moreover, the recycling of e-waste involves substantial costs and requires specialized techniques for sorting and processing [3]. Our study focuses on the separation and sorting of e-waste using machine learning and the recycling of plastic using pyrolysis, as well as the potential uses for the resulting bio-char by-product, and using metals to produce solar batteries. E-waste metals can be converted to solar batteries

for achieving sustainable and renewable energy sources, and we propose the use of time series data [4] for the continuous monitoring of the garbage level in the cloud, employing the Auto-regressive Integrated Moving Average (ARIMA) to forecast and analyze the life cycle [5].

Our system for collecting and sorting waste employs a combination of machine learning, cloud computing, and IoT technology, which streamlines the waste-to-asset process and centralizes it under a single sector. Our strengths in developing this system are convenience and efficiency in waste management; sustainability—by improving waste management and reducing the likelihood of overflowing bins, this system could help promote a more sustainable approach to waste disposal; and data collection and analysis—the system's ability to continuously update the trash level in the cloud and store data can provide insights into waste patterns. This helps inform waste management strategies. Our limitation for making this system is that difficulties, such as mode collapse, training instability, time series data and evaluating generated images may limit the GAN performance, while the quality of solar batteries and bio-fuels can vary due to impurities and chemical reactions, posing challenges for implementation in developing countries where establishing processes may be difficult.

This research revolves around addressing improvements in the efficiency of e-waste management. The primary objective is to explore and evaluate the feasibility and benefits of implementing IoT- and cloud-based smart systems in e-waste management processes, enabling seamless connectivity and communication between various devices and stakeholders involved in the e-waste management system. One of the focuses of our research is the utilization of machine learning algorithms for sorting e-waste. By using machine learning, our system will automatically identify e-waste. This not only saves labor in the sorting process but also enhances the accuracy and efficiency of recycling operations. The data-driven approach ensures the rapid collection of the e-waste, optimizes the utilization of available resources, enhances operational efficiency, and facilitates continuous improvement in e-waste management practices. By analyzing and interpreting relevant data, stakeholders can make informed decisions regarding waste collection, recycling methods, and resource allocation. In addition, our study describes how we can efficiently turn e-waste plastic into bio-fuel and bio-char. Over and above that, our research delves into the repurposing of e-waste metals for the production of solar batteries. With the ever-increasing demand for renewable energy sources, the conversion of e-waste metals into solar batteries offers a sustainable solution for both waste management and energy production. By utilizing these metals, it becomes possible to transform a potential environmental hazard into a valuable resource. Ultimately, the goal of this research is to contribute to a more sustainable and efficient e-waste management framework. By exploring the potential of IoT- and cloud-based systems, integrating machine learning techniques, investigating pyrolysis for recycling, repurposing e-waste metals for solar batteries, developing sustainable strategies, and promoting data-driven decision making, we can pave the way for a greener future and mitigate the environmental and health risks associated with e-waste.

The major contributions of this paper are summarized below:

- We have potentially enhanced the efficiency and accuracy of e-waste recycling processes, enabling a more effective sorting and separation of valuable components by leveraging these cutting-edge technologies.
- We have trained a digital image processing camera to recognize the perfect waste parts, and the interconnection between the sensors and cloud processes the data and recycles the wastes in a perfect manner.
- We have built a small prototype of an IoT-based waste management system with the help of the cloud, which will make this whole system automated.
- We are turning waste into assets by developing our waste management and recycling system.

This whole paper is organized in the following order: Section 2 provides the details of the related works; Section 3 provides information about our proposed system; Section 3.1 contains the proposed solution; Section 3.2 outlines the system architecture; Section 3.3 explains the methodology; Section 3.4 contains a flowchart; Section 3.5 shows the algorithm used; Section 4 contains performance analysis; Section 4.1 contains graphical analysis of e-waste level updates in the cloud; Section 4.2 shows an accuracy chart of the GAN algorithm; Section 4.3 contains graphical analysis of the pyrolysis method, Section 4.4 contains graphical analysis of solar battery production and the reduction in $CO_2$; Section 5 outlines limitations and future works; and Section 6 is our conclusion.

## 2. Related Works

The recycling industry faces a significant challenge in managing e-waste, as there is a pressing need to raise awareness among general people about the environmental and energy-saving advantages of recycling electronic devices. In Table 1 summary of related works are depicted. Addressing this challenge requires a comprehensive effort to educate users about the benefits of e-waste recycling [6]. Though it is a challenging task, it also presents notable opportunities to effectively navigate this complex field [7]. It requires a multifaceted approach that includes developing environmentally friendly products, effective waste collection, and safe and responsible recycling and disposal. Utilizing a magnetic field to segregate electronic waste into its constituent plastic and metal components represents a sophisticated approach. This method involves applying a magnetic field to the electronic waste, which causes the metallic components to be attracted to the magnet while the non-metallic plastic components remain unaffected [8]. Pyrolysis is an advanced technique that provides a sustainable and efficient solution for plastic parts of e-waste while also reducing the environmental impact of plastic waste [9]. The focus of the research paper [10] was to create an IoT-based monitoring system for e-waste, where they used microcontrollers and sensors to monitor e-waste. One effective strategy for reducing e-waste involves designing products with reusability in mind, inspiring creative reuse across different e-waste sources. Effective intervention strategies should aim to minimize exposure to toxic components in e-waste [11].

Bansod et al. [12] proposed a project that focuses on developing an IoT-based e-waste monitoring system. It utilizes ultrasonic sensors, an Arduino Mega 2560 microcontroller, and GSM communication to detect and monitor e-waste levels in real-time. The main benefits are efficient waste management, reduced overflowing bins, and improved planning of waste pick-ups. The limitations are a reliance on a 12 v source for the GSM module. This project's future work includes potential enhancements like incorporating a line follower robot for automated waste disposal. Bošnjakovic et al. [13] proposed a paper that examines the production of liquid fuel from plastic waste, focusing on technological, ecological, and economic aspects. Pyrolysis with a catalyst, particularly zeolite-based catalysts, is a well-established and mature technology for obtaining fuel from plastic waste. While up to 800 L of fuel can be obtained from one ton of waste plastic, real plants typically yield around 450 L. Disposing of waste plastics through fuel production offers significant environmental benefits, including reduced greenhouse gas emissions. However, waste separation, complex technical systems, and proximity to landfills for cost-effective transportation are

limitations.The analysis underscores the large amount of plastic waste in Croatia and the potential for economically viable bio-fuel production with improved waste collection.

**Table 1.** Summary of related works.

| Author and Year | Study Description | Limitations | Method Adopted |
|---|---|---|---|
| Bošnjaković et al. (2022) [13] | Technological and ecological dimensions of converting plastic waste into bio-fuel. | Sorting plastic from e-waste and the cloud; IoT use was not mentioned. | Pyrolysis to turn plastic waste into bio-fuel. |
| Devi et al. (2021) [9] | Emphasis on generating bio-fuels from plastic waste. | No discussion on IoT, the cloud, e-waste collection, or plastic-to-bio fuel conversion. | Process of proselytizing plastic waste into diesel fuel. |
| Shamsudin et al. (2022) [10] | IoT-based monitoring system using microcontrollers and sensors. | No discussion regarding the next steps after e-waste collection. | IoT-based project with microcontrollers and sensors. |
| Bansod et al. (2022) [12] | IoT-based system to detect e-waste. | Yet to implement a plan for utilizing the collected waste. | Monitoring garbage levels and communicating them through a GSM system. |
| Balakrishnan et al. (2015) [14] | Investigate the formation of bio-fuels from plastic scrap. | Generating bio-fuels; the methods for plastic collection are missing. | Pyrolysis to convert plastics to bio-fuels. |
| M H, Dinesh. (2020) [15] | Generate bio-oil using pyrolysis. | No mention of collecting plastic from e-waste or another place. | Thermal pyrolysis and catalytic pyrolysis. |
| kazi Shawpnil et al. (2023) [16] | QFD study conducted; combined efficient e-waste management methods. | No mention of the cloud, pyrolysis, bio-char, bio-fuel, or solar batteries. | Physical recycling for metallic parts, the biological method of mycoremediation, phytoremediation. |
| Abdullah Al Mamun et al. (2023) [17] | YOLOv5 to separate e-waste. | Pyrolysis, bio-char, bio-fuel, and solar batteries were not mentioned. | Pixy camera to recognize e-waste. |

In Table 2, a more in-depth analysis of related works is mentioned, where Sankeerth et al. [18] proposed a smart waste management system, which utilizes ultrasonic sensors in bins to measure garbage levels, which are then transmitted to a server via Wi-Fi. The server monitors the bins across the city, notifying the garbage truck driver when the amount of waste in a bin exceeds 70% and it needs to be emptied. SMS notifications are sent to the driver, providing optimized routes based on collected data. Thaseen Ikram et al. [19] proposes a waste management model for smart cities using a hybrid genetic algorithm (GA)–fuzzy inference engine. The system uses IoT components—RFIDs and sensors—to collect and process waste information. The model combines a GA with fuzzy logic to optimize the fuzzy inference system (FIS) and improve waste management accuracy. The system employs cost-effective sensors and ensures reproducibility. Their experimental results show high accuracy and precision of 95.44% in waste management and recyclable item classification. The proposed model reduces errors and minimizes manual interpretation costs compared to traditional approaches, but there could be potential privacy and security risks. Their future work includes integrating advanced technologies and addressing scalability and interoperability challenges. The smart dustbin proposed by Pavithra M. et al. [20] automatically opens upon detecting a clap or foot tap and closes once garbage is disposed of. An ultrasonic sensor monitors the garbage level and sends alerts to the main garbage collector when it reaches capacity. H. Cai et al. [21] proposed a garbage monitoring system where they use a NodeMCU chip integrated with ultrasonic sensors to measure waste levels in bins, which are transmitted to a cloud server through the Ali-cloud IoT platform. To observe the real-time bin status, they have used a web page. The average number of cleanings before establishing this was 3 and afterwards it was 2.28; the average number

of bin overflows before was 0.67 and afterwards it was 0.11, which added improvements in waste collection. The paper by Artang Sara et al. [22] introduces a tracking and tracing platform, which offers a user interface for users and administrators, providing essential information to users for the disposal of their e-waste. Integrating block-chain technology and circular economy approaches into the tracking platform and conducting comparative studies among different countries is their proposed future work.

**Table 2.** In-depth analysis of related works in relation to the benefits and risks of each approach.

| Reference Paper | Limitations | Method Adopted | Benefits | Risk | Future Work |
|---|---|---|---|---|---|
| [18] | Focuses on data collection and monitoring without data analysis for process optimization. | Bins with ultrasonic sensors measure garbage levels, send data to a server via Wi-Fi, and optimize collection routes using SMS. | Direct message sending reduces the costs and maintenance for the embedded bins, enhancing independence and transparency. | The reliance on Wi-Fi and server stability poses a risk of data loss and potential failure in communication. | Incorporating a database and utilizing data analytics to optimize waste management processes and improve efficiency |
| [19] | Limited applicability may impact its practicality in different waste management contexts. | An IoT and fuzzy inference system with a genetic algorithm to create a waste disposal system | Enhanced waste management efficiency, cost reduction, and resource optimization | Reliability, accuracy, privacy, and security | The integration of additional advanced technologies, scalability, and interoperability |
| [20] | Foul odors emanating from the bins and the manual control of the dustbins can restrict mobility and flexibility in waste collection. | Uses sensors for gesture detection and garbage level monitoring, enabling automatic bin operation and timely emptying through an IoT web interface | The automated waste management system reduces labor costs and enables the timely disposal of garbage to the correct location. | The system relies on the accuracy and reliability of the sensors to detect the garbage levels. False readings could impact the efficiency. | The system contributes to waste reduction, resource conservation, and sustainable waste management by handling both metal and non-metal waste. |
| [21] | Alerts cleaners based on threshold parameters but does not differentiate the recycling of e-waste | A sensor-based device detects and monitors the garbage status and sends notifications to cleaners when thresholds are exceeded. | The notification system eliminates the need for continuous monitoring, as it alerts the cleaner when the dustbin requires cleaning. | The device might get damaged while using the dustbin as it is set in quite an unprotected manner. | An Android app will be developed for this in future and a better algorithm will be implemented here. |
| [22] | The limited waste registration and complex interface may impede tracking and user adoption in the application. | E-waste registration, QR code tracking, and Google API integration enable effective monitoring and proper disposal. | The stakeholders can track e-waste and its location effectively through unique identification, facilitating easy monitoring. | The only risk is the security issue as the users are giving certain mobile access via this application. | Create a sustainable system with robotics and blockchain for enhanced security and tracking capabilities. |

## 3. Proposed System

### 3.1. Proposed Solution

Our main goal is to collect e-waste and send it for recycling in an efficient and automated manner. We are using the combination of the IoT and machine learning for gathering e-waste for recycling purposes. We will be placing the processing part of our system in a dumpster with the help of a Field-Programmable Gate Array (FPGA) using the GAN algorithm to distinguish the e-waste from other wastes. Our proposed solution entails the deployment of a smart bin to collect waste, which utilizes cloud-based technology

to monitor and update the garbage level automatically. If the bin reaches its maximum capacity, the SIM900A module generates a message alerting the collectors. Upon collection, we implement a process to separate the metallic and plastic components of the waste. The plastic components undergo a pyrolysis process to yield bio-fuel, while the metallic components are repurposed for solar panel and battery production.

### 3.2. System Architecture

The proposed system architecture is depicted in Figure 1. The system includes several steps aimed at effectively managing e-waste. The initial phase (step-1) involves the classification of wastes based on their type, which will be conducted by machine learning. Subsequently, e-waste is collected and deposited into a smart bin, and based on the trash level data, a data-driven decision-making process is implemented in Figure 2 to determine whether a notification should be sent to the trash collector. This process involves evaluating the trash -level data against predetermined thresholds, and if the data exceeds these thresholds, a notification is triggered and sent to the trash collector in step-2 and step-3.

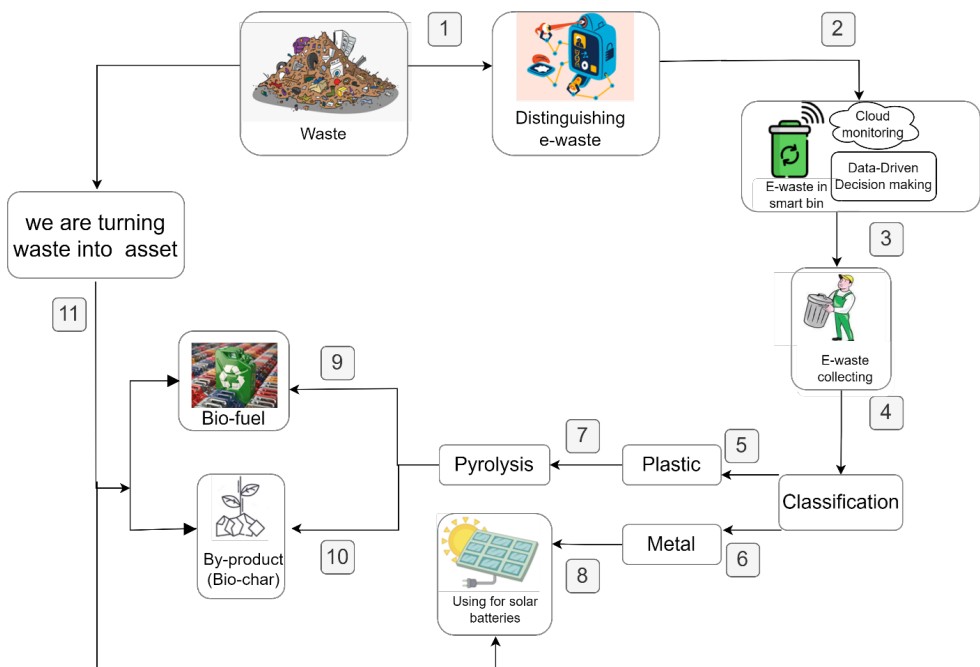

**Figure 1.** System architecture of our proposed solution.

The cloud-based system is continuously monitoring the level of trash in the background. In step-4, the e-wastes are separated into two categories, plastic and metal. The metal waste is processed for solar batteries in step-8, while plastic waste is converted into bio-fuel using the pyrolysis process and we obtain bio-char as a by-product in step-9 and step-10. In the final step of the process, the repurposed and transformed wastes are converted into valuable assets.

Figure 2 illustrates a data-driven decision-making process for e-waste collection. Trash level data are continuously monitored using an ultrasonic sensor in the trash bin. This data are collected, enabling real-time analysis of e-waste levels. Based on the analysis, notifications are sent to e-waste collectors, prompting them to collect e-waste from specific bins. The collectors follow the notifications, collect the e-waste, and ensure proper recycling methods are employed. This data-driven process optimizes the collection efficiency and helps in the timely and targeted collection of e-waste, contributing to environmental sustainability and proper e-waste management.

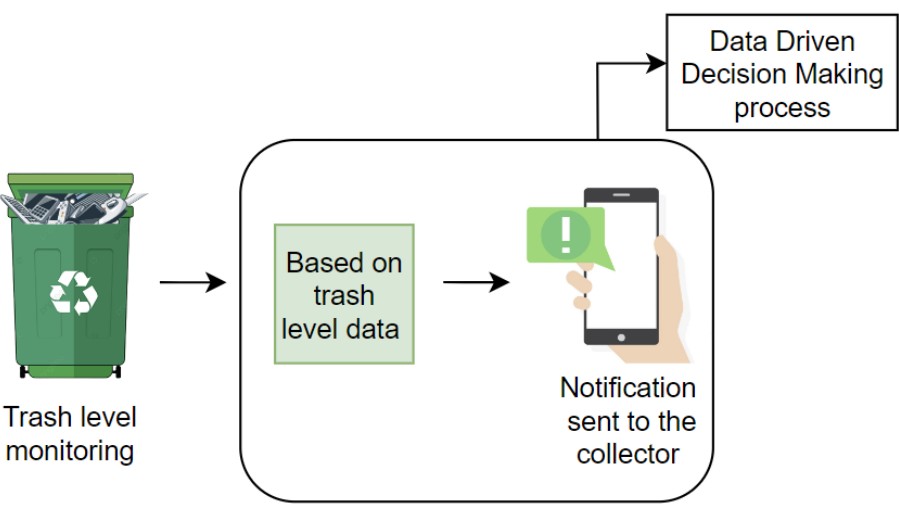

**Figure 2.** Data-driven decision making process.

Figure 3 is depicting three layers [23]. The sensor layer consists of a smart dustbin with an ultrasonic sensor that detects the level of trash inside the bin. The data collected from the sensors are sent to the cloud layer using the ESP-8266 Wi-Fi Module. The cloud layer receives the data from the sensor layer and stores it in a time series database. A time series database is designed to handle data that are collected over time, such as the trash level in the bin. The data stored in the time series database can be queried and analyzed to generate insights and predictions about the future. The Auto-regressive Integrated Moving Average algorithm is applied to the database to forecast the trash level for the future. The cloud layer also provides an interface for the user to view the trash level and other information in real time and send the value in the microcontroller. The user can access this interface through a web or mobile application. If the level of trash in the bin reaches a certain threshold, the microcontroller sends the notification using the GSM module to the application layer. The application layer receives the notification, and the collector collects the data after receiving it. The smart dustbin system uses a combination of sensors, cloud computing, and predictive algorithms to collect and analyze data about the trash level in the bin. These data are used to provide real-time notifications to the user and improve waste management processes.

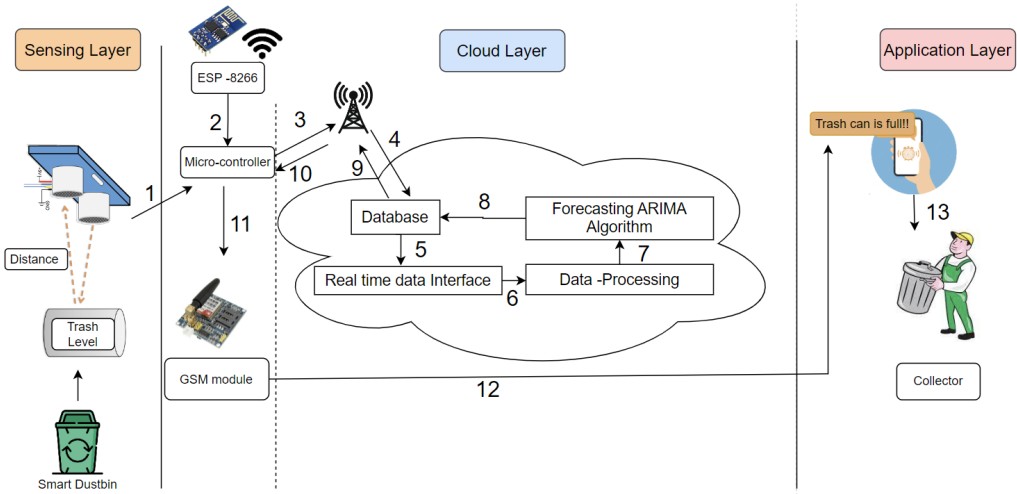

**Figure 3.** System architecture of collecting and monitoring trash using cloud and IoT.

*3.3. Methodology*

Our working process entails a dustbin for e-waste. The waste will move along a conveyor belt, and in the initial section of the belt our processing part will be incorporated, featuring a trained camera with machine learning and a Generative Adversarial Network (GAN) algorithm. The GAN requires high computation power and memory resources [24]. A powerful, dedicated hardware platform such as FPGA provides the high computation power. There are two sections to the GAN, namely the discriminator and the generator.

We provide the dataset to the discriminator, and the generator monitors the waste and creates an image according to it. The generator and discriminator images are then compared, and if they match the waste will be thrown into the e-waste smart bin, while other waste will be deposited in a different pile. The smart bin contains an ultrasonic sensor, which sends waste-level data to the database in the cloud through an ESP8266 Wi-Fi module; each data point that is sent from the sensor to the time series database should include a timestamp and the trash level reading in Figures 6–8. The time series database will store these data and make them available for querying and analysis. To forecast the trash level, we are using the ARIMA model [25]. The forecasting can be used to optimize the system by predicting when the trash level will reach the threshold and scheduling pick-ups accordingly. This can help reduce costs and improve efficiency. If the garbage level exceeds a threshold value, a notification will be sent to the collector via the SIM900A GSM module, and it will be sent for recycling. In the recycling process, the e-waste is churned through a robust blade and separated into plastic and metallic parts using a magnetic field. The plastic parts will be sent for the pyrolysis process. In pyrolysis there are several steps:

As the plastic is already shredded, it will increase the surface area for improved pyrolysis. Shredded plastic is fed into a pyrolysis reactor. Pyrolysis is a type of thermal treatment that breaks down complex organic materials (plastic) into simpler compounds using heat in the absence of oxygen. The end products of pyrolysis are typically a liquid fuel known as pyrolysis oil or bio-oil and a gaseous mixture known as syngas, which can be used for various applications, such as energy generation and chemical production [26]. Since pyrolysis takes place in the absence of oxygen, the reactor is sealed to prevent air ingress. The reactor is heated to an elevated temperature. The temperature and pressure inside the reactor are carefully controlled to ensure that the plastic is efficiently converted into bio-fuel. When plastic is heated, it begins to decompose into components, such as gas, liquid, and char. Gases and liquids are condensed and collected as bio-fuel.

Once the pyrolysis process is complete, the reactor is cooled and the bio-fuel is recovered from the condenser. The collected bio-fuel may require further purification to remove impurities such as water and acids. This can be performed by methods such as filtration or distillation. Finished bio-fuel products are stored in tanks or containers until use. If any organic material is mixed with the plastic waste, such as bio-solids, the by-products of pyrolysis, bio-char, can be recycled and used in various applications, such as soil amendment, carbon sequestration, and energy production [15]. This product has shown notable advantages in eliminating pollutants from wastewater [27] and enhancing soil quality [28]. Moreover, we are using scrubber and electrostatic precipitators to control pollution [29]. Metallic waste can be used for making solar batteries. The process of converting metal churns from e-waste into solar batteries involves several steps: The shredded metal is treated with acid or other chemicals to extract impurities and separate the pure metals. The pure metals are then processed using electrolysis, which involves passing an electric current through a solution containing the metal ions. This process causes the metal ions to gain or lose electrons, resulting in the formation of metal deposits on electrodes. The metal deposits are then used to produce various components of a solar battery, including the anode, cathode, and electrolyte. These components are assembled to create a functional solar battery that can store and release energy. The exact process of converting metal churns into solar batteries can vary depending on the specific type of metal and the desired end product. However, in general, the process involves a combination of chemical and electro-chemical techniques to purify and refine the metal and then assemble it into battery components.

The technical aspects of our methodology are as follows:

Data collection method:

Camera and machine learning: trained camera system with machine learning capabilities was used to monitor and capture images of the waste on the conveyor belt.

Ultrasonic sensor: the e-waste smart bin was equipped with an ultrasonic sensor to measure the waste level.

ESP8266 Wi-Fi module: the waste level data from the ultrasonic sensor were transmitted to a cloud database using an ESP8266 Wi-Fi module.

SIM900A GSM Module: when the garbage level exceeded the threshold, a notification was sent to the collector via a SIM900A GSM module.

Data analysis techniques: the ARIMA model was employed for forecasting the trash level.

*3.4. Flowchart*

In Figure 4, the flowchart describes the process of our IoT- and cloud-based e-waste management, starting with the aggregation of various types of waste. We utilize a trained camera, which has been trained with a GAN algorithm, for the classification of e-waste. Through image processing, it determines whether the waste is e-waste or not. If it is not e-waste, it is dumped in a different pile; otherwise, it is deposited in the smart bin. As e-waste is being disposed of, the waste level continues to increase; this increased level data are then updated in the cloud, and the system checks if the bin is full through an ultrasonic sensor. If it is not full, the process continues, or else a notification is sent to the collector. After collecting the e-waste, the recycling steps begin. It starts with churning the e-waste, followed by magnetic field separation. From separation, there are two parts—plastic and metallic churns. Plastic goes through pyrolysis and becomes bio-fuel, while metallic churns are processed for solar batteries. The process ends with the production of bio-fuel, with bio-char as a by-product, and solar batteries, representing our system's effective transformation and the recycling of e-waste into sustainable and eco-friendly materials.

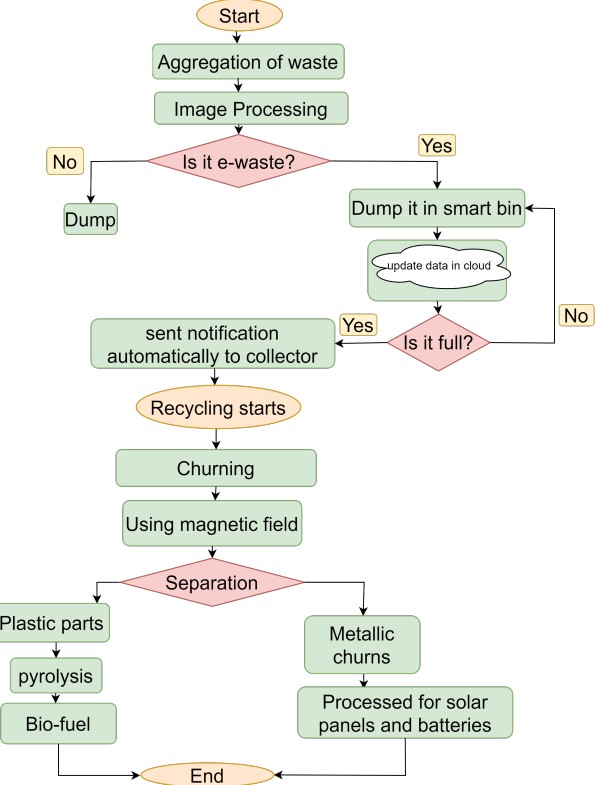

**Figure 4.** Flowchart of Proposed System.

*3.5. Algorithm*

For image processing, we are using the GAN, which is a very high-level algorithm. The accuracy of GAN algorithms for image processing is highly dependent on the specific use case and the techniques employed to train and optimize the model. Here, using GAN, the machine will be trained with real-life e-waste images and with the help of those images it will provide its decision. The pseudo code of the GAN algorithm is given below:

In Algorithm 1, the generator network G and discriminator network D are initialized with random weights. The hyper-parameters $\alpha$, $\beta$, and $\gamma$ are initialized. For a specified number of training iterations, the following steps are executed. For a specified number of discriminator updates per generator update, the following steps are executed. A mini-batch of m real images from the dataset is sampled. A mini-batch of m noise samples from a noise distribution is sampled. Fake images are generated using the generator network G. The discriminator network D is updated by minimizing the binary cross-entropy loss between the real images and the fake images, with the gradients computed using back-propagation. A mini-batch of noise samples from a noise distribution is sampled. The generator network G is updated by taking a gradient step on the loss function that maximizes the binary cross-entropy loss between the generated images and the real images, with the gradients computed using back-propagation. The hyper-parameters $\alpha$ and $\beta$ are updated using a decay factor $\gamma$. The trained generator network G is returned.

---

**Algorithm 1** Image classification using Generative Adversarial Networks

---

1: Initialize the generator network $G$ with random weights
2: Initialize the discriminator network $D$ with random weights
3: Initialize the hyper-parameters $\alpha$, $\beta$, and $\gamma$
4: **for** number of training iterations **do**
5:     **for** number of discriminator updates per generator update **do**
6:         Sample a mini-batch of $m$ real images from the data-set
7:         Sample a mini-batch of $m$ noise samples from a noise distribution
8:         Generate fake images using the generator network
9:         Update the discriminator network.
10:     **end for**
11:     Sample a mini-batch of noise samples from a noise distribution
12:     Update the generator network by taking a gradient step on the loss function
13:     Update the hyper-parameters
14:         $\alpha \leftarrow \gamma\alpha$
15:         $\beta \leftarrow \gamma\beta$
16: **end for**
17: **return** the trained generator network $G$

---

We use generator and discriminator neural networks to train on a dataset of real e-waste images. The goal is to train the generator network to produce images that are indistinguishable from real images, while the discriminator network learns to distinguish between real and generated images. During training, the generator produces images to try and fool the discriminator, and the discriminator tries to become better at distinguishing between real and generated images. Once trained, the generator can generate new images, which can be evaluated by comparing them to real images. If they are similar, the waste can be disposed of in the appropriate destination dustbin.

Algorithm 2 is used to detect the level of e-waste in a smart dustbin and send a notification to the garbage collector when the dustbin is almost full. The input variables for this algorithm are *n* (the number of iterations), *x* (the echoP input), and *y* (the trigP input). The algorithm starts by initializing *x* and *y* with the echoP and trigP inputs, respectively. The threshold distance is set to 4, which is the maximum distance at which the smart dustbin can detect e-waste. The algorithm then puts e-waste in the smart dustbin and enters a loop that runs as long as n is not equal to 0. Inside the loop, the value of *y* is set to

0 or Low initially. Then, the algorithm runs for 10 iterations, and the value of $y$ is set to 1 or High. After 10 iterations, $y$ is again set to 0. The value of $x$ is set to 1 or High.

The algorithm then calculates the distance between the smart dustbin and the e-waste using time and the speed of sound. The garbage level is calculated by subtracting the distance from the total dustbin distance. This information is sent to the cloud using ESP8266. If the distance is greater than or equal to the threshold distance, the algorithm sends a notification using the SIM900A GSM module to the garbage collector. Otherwise, the smart dustbin collects the e-waste.

---

**Algorithm 2** An algorithm for calculating the e-waste level in a smart dustbin

---

**Require:** $n \geq 0$
  $x = echoPin$
  $y = trigPin$
  $n = iteration$
  $thresholdDistance = 4$
  Put e-waste in Smart dustbin
  **while** $n \neq 0$ **do**
     $y \leftarrow 0$ or Low
     **for** number of 10 iterations **do**
        $y \leftarrow 1$ or High
     **end for**
     $y \leftarrow 0$
     $x \leftarrow 1$ or High
     $distance \leftarrow \frac{time \times 0.034}{2}$
     $garbageLevel = totalDustbinDistance - distance$
                                                    ▷ Send the distance and garbage level in cloud
     **if** $distance \geq thresholdDistance$ **then**
        Sent notification to garbage collector
     **else**
        Smart dustbin collects the e-waste
     **end if**
  **end while**

---

## 4. Performance Analysis

### 4.1. Graphical Analysis of E-Waste Level Updates in Cloud

Figure 5 depicts the updates of the garbage level in the cloud of a certain time period, where the initial level (at time = 1) was recorded as 28 cm, indicating an empty smart dustbin. As observed from the time axis (y-axis), the garbage level gradually decreased until it reached 9 cm at time = 8. At time = 9, the garbage level reached a threshold distance of 4 cm, after which it remained constant until time = 12. During this period, a notification was sent to the collector, who subsequently collected and emptied the e-waste from the smart bin. Following the trash collection, the garbage level increased and was recorded as 28 cm at time = 13.

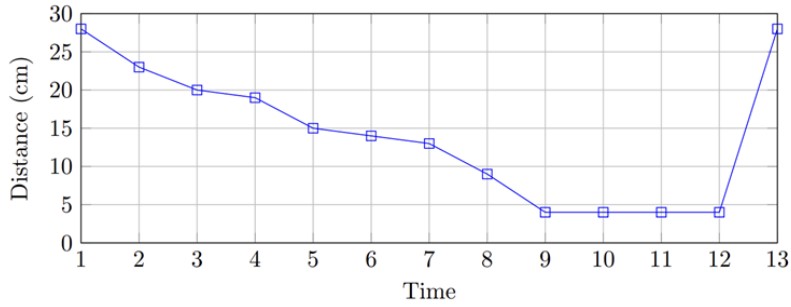

**Figure 5.** Graphical analysis of e-waste level update.

Figures 6–8 illustrate the empty space available in the smart trash bin and the process of updating the corresponding values in the cloud for a certain period of time. The distance between the contents and the top of the trash bin indicates the level of empty space available. A greater distance corresponds to a higher amount of empty space, while a lesser distance corresponds to a lower amount of empty space.

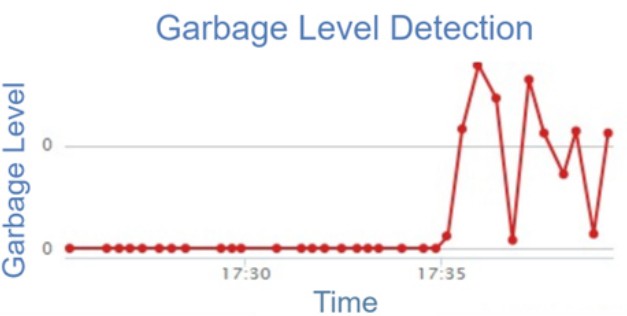

**Figure 6.** E-waste level update information in cloud.

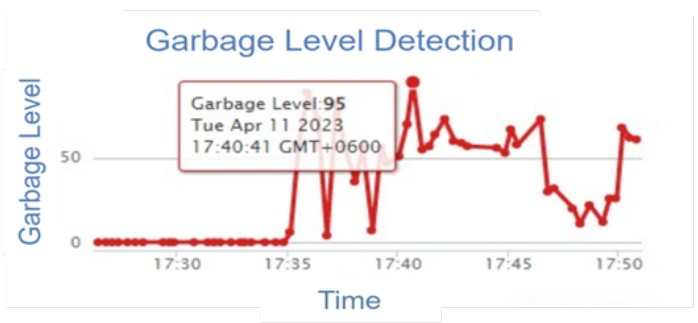

**Figure 7.** E-waste level update information in cloud with timestamp.

```
Content-Length: 30 Field-1: 37 data send
Waiting
Distance =29 cm
Send data to thingspeak:
Content-Length: 30 Field-1: 29 data send
Waiting
Distance =20 cm
Send data to thingspeak:
Content-Length: 30 Field-1: 20 data send
Waiting
Distance =10 cm
Send data to thingspeak:
Content-Length: 30 Field-1: 10 data send
Waiting
Distance =6 cm
```

**Figure 8.** E-waste level update information in cloud showing in serial monitor.

### 4.2. Accuracy Chart of GAN Algorithm

We are using the GAN algorithm where we are dividing our dataset into training, validation, and testing sets. The training set is used to train the model, the validation set is used to tune the model's hyper-parameters, and the testing set is used to evaluate the model's performance on unseen data.

This accuracy chart in Table 3 shows the performance of an e-waste recognition system that uses the GAN algorithm. The chart shows the precision, recall, and F1-score for each category of e-waste that the system is designed to recognize: smartphones, laptops, televisions, monitors, and other, which includes all other types of e-waste.

**Table 3.** Accuracy Chart for E-waste Recognition System using GAN Algorithm.

| Category | Precision (%) | Recall (%) | F1-Score (%) |
|----------|---------------|------------|--------------|
| Phone | 95 | 97 | 96 |
| Laptop | 90 | 85 | 87 |
| TV | 85 | 91 | 88 |
| Monitor | 92 | 89 | 90 |
| Other | 80 | 75 | 77 |
| Overall | 90 | 88 | 89 |

In Figure 9, precision is a performance metric that measures the accuracy of a system in identifying relevant items. It quantifies the proportion of correctly identified items among the total items identified by the system. The precision is calculated as the ratio of true positives (items correctly identified) to the sum of true positives and false positives (items incorrectly identified). A high precision value indicates that the system is effective in accurately identifying relevant items. It signifies that the system has a low rate of falsely identifying unrelated items as the target item. In our example, a precision of 95% implies that the system has a relatively low rate of falsely identifying non-smartphone items as smartphones.

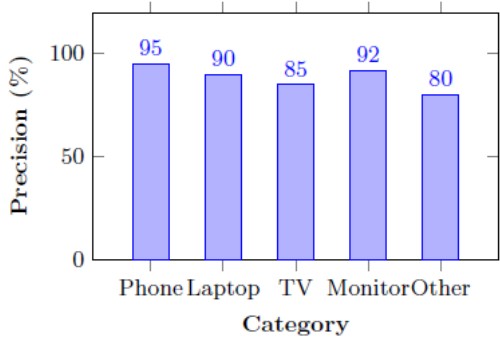

**Figure 9.** Precision of each category.

However, it is important to note that precision alone may not provide a complete picture of the system's performance. It should be considered in conjunction with other performance metrics, such as recall and the F1-score, to have a comprehensive evaluation of the system's effectiveness in identifying relevant items. This was considered in Figure 12. For instance, let us consider the example of a system that identifies smartphones among various items. The precision of the system in identifying smartphones is 95%, which means that out of all items identified as smartphones, 95% were actually phones. The remaining 5% could be items incorrectly classified as smartphones.

In Figure 10, recall serves as a performance metric that quantifies the completeness or comprehensiveness of a system in identifying relevant items. It is calculated by dividing the number of true positives (correctly identified items) by the sum of true positives and false negatives (items that were not identified as belonging to a particular category but should have been). Recall is particularly significant in situations where the consequences of false negatives are critical. By achieving a higher recall value, the system minimizes the chances of overlooking relevant items and provides a more comprehensive identification process.

For instance, consider a system designed to identify laptops among various objects. If the system achieves a recall of 85%, it indicates that out of all the actual laptops in the sample, 85% of them were correctly identified by the system. The remaining 15% represents the instances where the system failed to recognize laptops that were present.

A higher recall value suggests that the system is effective in capturing a larger proportion of the relevant items. It indicates a lower rate of false negatives, meaning that fewer items belonging to the target category are missed by the system. In our example, a recall of

91% signifies that the system has a relatively high ability to detect and include televisions in its identification process.

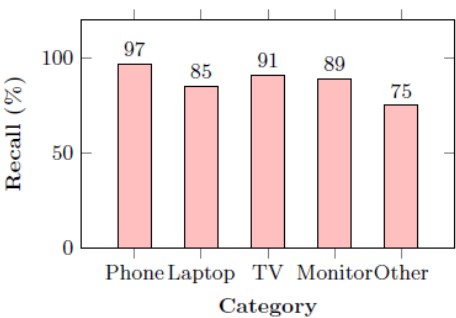

**Figure 10.** Recall of each category.

In Figure 11, the F1-score is a metric that encompasses both precision and recall to provide a comprehensive evaluation of the performance of a classification model. Precision and recall are both crucial aspects in assessing the effectiveness of such models. The F1-score offers a balanced measure by taking the harmonic mean of precision and recall.

$$F1\text{-}score = \frac{2 \cdot (precision \cdot recall)}{precision + recall}$$

**Figure 11.** F1-Score (%) of each category.

This choice is made because the harmonic mean assigns more weight to smaller values, ensuring that both precision and recall are given equal consideration. By considering both precision and recall in the F1-score, it provides a unified indicator of overall performance. It strikes a balance between the two metrics, giving equal importance to correctly identifying relevant items (precision) and capturing the full extent of relevant items (recall). The F1-score is particularly valuable when the class distribution is imbalanced or when both precision and recall are of equal importance. It offers a single value that represents the overall effectiveness of the classification model, allowing for easier comparison and decision making.

In Figure 12, the overall performance of the system is represented by the "Overall" row of Table 3. Here, P represents precision, R represents recall, and F represents the F1-Score. This row displays the key performance metrics, including the precision, recall, and F1-score. According to the table, the system achieves an overall precision of 90%, recall of 88%, and F1-score of 89%. These metrics provide a comprehensive assessment of the system's performance across all categories. With a precision of 90%, the system demonstrates a high level of accuracy in correctly identifying e-waste items. Similarly, the recall of 88% indicates that the system is effective in capturing a significant portion of the actual e-waste items present in the sample.

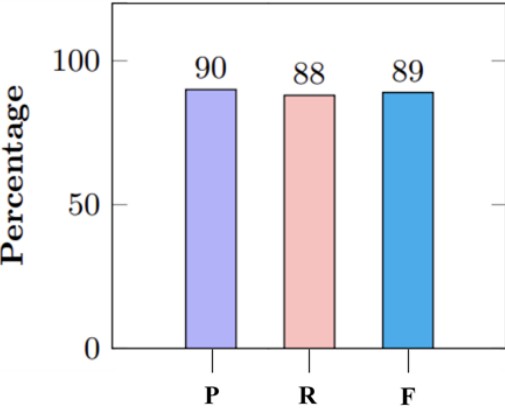

**Figure 12.** Overall performance of each category.

The F1-score of 89% is a balanced measure that combines precision and recall. It considers both metrics to provide an overall evaluation of the system's performance. This score indicates that the system maintains a good balance between precision and recall, achieving a harmonious trade-off between accurately identifying e-waste items, but may have some minor errors in specific categories.

*4.3. Graphical Analysis of Pyrolysis Method*

Figure 13 shows the yield of bio-fuel from plastic waste using the pyrolysis method. The x-axis represents the temperature in degrees Celsius, while the y-axis represents the yield of bio-fuel as a percentage. The blue line shows the relationship between the temperature and bio-fuel yield. As the temperature increases, the yield of bio-fuel also increases. At a temperature of 300 °C, the yield is 20%, which increases to 50% at a temperature of 500 °C. This graph suggests that the pyrolysis method can be an effective way of producing bio-fuel from plastic waste and that higher temperatures can result in a higher yield of bio-fuel. The legend indicates that the red line represents the bio-fuel yield.

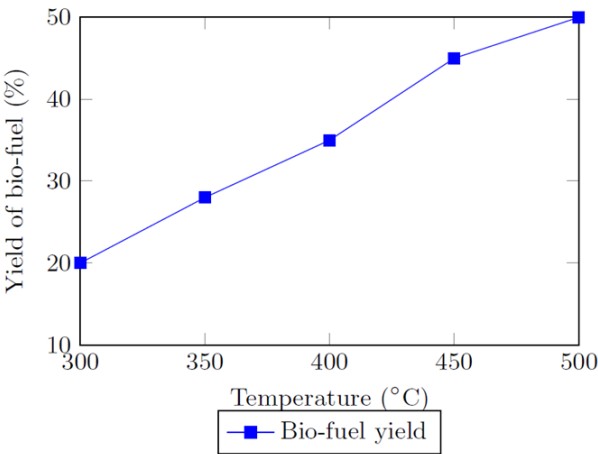

**Figure 13.** Yield of bio-fuel from plastic waste using pyrolysis method.

Table 4 [30] displays the results of the elemental analysis of mixed waste plastic pyrolysis liquid samples obtained from both thermal pyrolysis and catalyzed pyrolysis processes. The table shows the weight percentages of carbon (C), hydrogen (H), nitrogen (N), and sulfur (S) in the samples. The results indicate that the catalyzed pyrolysis process had a higher percentage of carbon and a lower percentage of hydrogen compared to the thermal pyrolysis process. Additionally, both processes showed similar percentages of nitrogen and sulfur in the pyrolysis liquid samples.

**Table 4.** Elemental analysis of mixed waste plastic pyrolysis liquid samples.

| Weight (%) | Thermal Pyrolysis | Catalysed Pyrolysis |
|:---:|:---:|:---:|
| C | 94.24 | 97.11 |
| H | 11.73 | 10.12 |
| N | 0.61 | 0.28 |
| S | 4.8 | 4.36 |

*4.4. Graphical Analysis of Solar Battery Production and Reduction in $CO_2$*

According to a study conducted in the Bangladesh University of Engineering and Technology, the projected growth of e-waste in Bangladesh from 2010 to 2035 is expected to increase from 0.13 million tons in 2010 to 4.62 million tons by 2035, indicating a significant rise in electronic waste generation over the given time period. The recycling rate of e-waste in the country stands at a mere 3%, with the remaining majority being indiscriminately deposited in landfills and rivers [31]. For each ton of e-waste that is collected and recycled, an impressive 1.44 tons of $CO_2$ emissions are effectively circumvented, as per the findings of an in-depth analysis conducted by the esteemed Belgian $CO_2$ environmental consulting firm, $CO_2$ logic [32].

Figure 14, shows the amount of $CO_2$ emissions in million tons through over the years if we recycle the metal from e-waste perfectly and reuse it for making solar batteries. The red line represents $CO_2$ emissions without recycling e-waste at all in Bangladesh and the blue line represents them after recycling only 3% of e-waste [31]. The green line represents the amount of $CO_2$ reduction if we recycle 60% of e-waste in Bangladesh. It is clearly visible that if we recycle the e-waste and produce solar batteries we will be able to reduce $CO_2$ emission to a large extent.

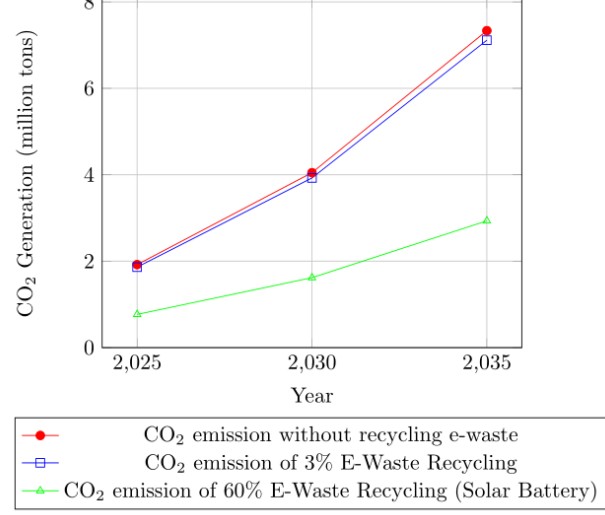

**Figure 14.** Reduction in $CO_2$ emissions with metal recycling from e-waste for solar batteries.

**5. Limitations and Future Works**

We have added the limitations and future works of our research:

The pyrolysis plant structure may vary according to its feedstock, requirements, products, and specific requirements. In conformity with HUAYIN, a manufacturer of waste tyre/plastic pyrolysis plants [33] typical a pyrolysis plant consists of six primary systems. The emission control system purifies the gas and confirms the emission of clean air. It also offers various de-dusters as per customer necessity while maintaining the standard of qualified emission. Pyrolysis is considered the future for plastic recycling techniques. We will implement preventive measures to mitigate and minimize the impacts of pyrolysis, such as the following:

- **Emissions and potential health/environment risks:** Pyrolysis can release gas, volatile organic compounds, and toxic substances, but our proposed process includes the implementation of a proper emission control system.
- **Energy inputs:** In some cases, traditional pyrolysis may use fossil fuels; our research focuses on using renewable energy sources as this will minimize the carbon footprint.
- **Contaminant release:** Plastic parts of e-waste can contain contaminants. Our proposed pyrolysis process involves control measures to ensure the safe handling, advanced sorting, and treatment of the e-waste plastic.

The proposed system will take every measure to reduce all the possible detrimental effects of pyrolysis. The major limitations of our system might be as follows:

- **Feedstock variability:** E-waste plastics may contain a different range of materials with different compositions. The consistency of the pyrolysis process can face some hurdles due to this.
- **Contaminants and impurities:** Despite thorough sorting and taking advance measures, some contaminants may still be present in the feedstock, which can affect the quality of the whole process and demean the standard of the by-products.
- **Pollutant emissions:** Though we are using an emission control system, making efforts to minimize the emissions, a comprehensive understanding about all kinds of pollutants and their potential impacts can impose limitations.

Continuous experimentation, development, and further research are necessary to enhance the characterization and monitoring of emissions, ensuring the safety and environmental sustainability of e-waste pyrolysis. For the future work of our research paper, we want to include some aspects: the further optimization of the pyrolysis process; the enhancement of data-driven decision making by leveraging advanced technologies; waste stream analysis for an effortless recycling process; solar batteries management and control to optimize their performance and prolong their lifespan; and the optimization of the recycling workflow to train the recyclers for different recycling processes.

### 6. Conclusions

The IoT- and cloud-based waste management and recycling system we have implemented successfully addresses the pressing issue of e-waste. Our study focused on the efficient separation and quick disposal of e-waste using the IoT, cloud computing, and machine learning. Our research results showcased numerous advantages, including enhanced efficiency, cost reductions, improved monitoring capabilities, and increased sustainability. Real-time data collection and analysis facilitated optimized waste collection routes, minimized the environmental impact, and successfully produced bio-fuel and solar batteries. Our research objectives were achieved through the implementation and evaluation of an IoT- and cloud-based waste management system, resulting in improved waste monitoring, optimized collection routes, and turning waste into assets by producing bio-fuel through pyrolysis and converting e-waste metal into solar batteries. Our study's outcomes align seamlessly with our initial research objectives, demonstrating the system's ability to overcome challenges associated with traditional waste management practices. However, there are some limitations, such as security and privacy concerns related to IoT devices and cloud infrastructure that must be addressed with robust measures to ensure data protection. In addition, the performance of the GAN algorithm can be affected by issues such as mode collapse, where the generator produces limited varieties of output, instability during training, and difficulty in evaluating the generated images. To sum up, the implementation of our IoT- and cloud-based waste management system has immense potential to revolutionize waste management practices. Its real-time data gathering, operational optimization, resource allocation, and production of recycled products offer substantial cost savings, a reduced environmental impact, and improved sustainability. However, addressing security concerns and conducting further research to ensure widespread adoption are necessary tasks for the successful implementation of such systems in the future.

**Author Contributions:** Conceptualization, M.F. and A.B.F.; methodology, M.F., A.B.F. and S.E.A.; software, M.F., A.B.F., S.E.A. and M.M.I.; validation, M.F., A.B.F., S.E.A. and M.M.I.; formal analysis, M.F. and A.B.F.; investigation, M.F., A.B.F., S.E.A. and M.M.I.; resources, M.F., A.B.F., S.E.A. and M.M.I.; data curation, M.F., A.B.F., S.E.A. and M.M.I.; writing—original draft preparation, M.F. and A.B.F.; writing—review and editing, M.F., A.B.F., S.E.A. and M.M.I.; visualization, M.F., A.B.F. and S.E.A.; supervision, M.F. and M.M.I.; project administration, M.F., A.B.F. and M.M.I. All authors have read and agreed to the published version of the manuscript.

**Funding:** This research received no external funding.

**Data Availability Statement:** Publicly available data-set were analyzed in this study. This data-set can be found here: [https://universe.roboflow.com/new-workspace-f7og7/e-waste-mx8fq/dataset/1].

**Conflicts of Interest:** The authors declare no conflict of interest.

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
