# Peer review of "An IoT- and Cloud-Based E-Waste Management System for Resource Reclamation with a Data-Driven Decision-Making Process"

_2624-831X, doi:10.3390/iot4030011_

Round 1

Reviewer 1 Report

I had the pleasure of thoroughly reviewing the paper titled "An IoT-Cloud based E-waste management system for resource reclamation with a data-driven decision-making process." The topic is interesting and important in the current context. I found the paper to be well-structured, presenting clear ideas that were easy to follow. The background section provided a comprehensive summary of the main research in the field, and the references are new and appropriate. Moreover, the results were well presented and effectively supported by thorough research. However, I have the following observations. Firstly, the research questions are missing. It is important to explicitly formulate them to help readers understand the specific goals of the research. Additionally, the methodology section primarily describes the physical process rather than the research approach. Both aspects are important, and I recommend including details about the research methodology alongside the technical aspects. Furthermore, more details about figure 4 are needed to enhance its comprehensibility. The resolution of figure 3 could be improved. Lastly, expand on the conclusion section by providing a more detailed summary of the main findings and their implications. Discuss how the results align with the research objectives and consider addressing any limitations or areas for future research. By incorporating these suggestions, you can enhance the overall clarity and completeness of the paper, making it more impactful and valuable to readers.

Author Response

Please find the review responses in the attached file.

Reviewer 2 Report

The leverage of IoT and cloud technologies in e-waste management has huge potential - it can increase collection efficiency, improve recycling processes, reduce environmental impact and promote a circular economy approach to e-waste.

The proposed smart e-waste management system appears to be highly efficient, especially in terms of component recognition (recall and F1-score is relatively high).

Hence, I perceive the proposed model as successful. What concerns me, however, is the further processing of e-waste.

My concerns relate to these, in particular the processing of plastic recyclate into biochar:

- Pyrolysis releases gases, including volatile organic compounds (VOCs) and other potentially toxic substances. If not properly controlled, these emissions can pose health and environmental risks.

- The pyrolysis process requires significant energy inputs, often from fossil fuels. If renewable energy sources are not used, this can contribute to greenhouse gas emissions and exacerbate climate change.

- Plastic waste may contain various contaminants such as heavy metals, additives or other non-plastic materials. These contaminants can be released during pyrolysis and potentially contaminate the resulting biochar, limiting its usability and posing a risk to soil and water quality.

- What is the economic viability and cost-effectiveness of pyrolysis technology for converting e-waste plastics to biochair?

Moreover, although pyrolysis can convert plastic waste into biochar, it is generally considered preferable to reduce, reuse and recycle plastic waste whenever possible. Pyrolysis of plastic waste should be considered as a last resort, as it is preferable to give priority to strategies that avoid plastic waste and promote the use of environmentally friendly alternatives.

Please address these issues and problems consistently.

The summary of related works should be expanded to include current studies on e-waste management.  A more in-depth analysis of these works in relation to the benefits and risks of each approach should be provided.

Figure 14 needs more consistent explanation.

Author Response

(The authors gave the same response as above.)

Round 2

Reviewer 2 Report

Dear Authors, thank you very much for considering my comments and addressing my concerns. I still have concerns regarding the pyrolysis process. The reference you are citing is for the pyrolysis of biomass, not e-waste. For me, this is not sufficient argumentation. I require you to add a section devoted to the limitation of your research, where you will address this issue properly.
